# Different Approximation Methods for Calculation of Integrated Information Coefficient in the Brain during Instrumental Learning

**DOI:** 10.3390/brainsci12050596

**Published:** 2022-05-03

**Authors:** Ivan Nazhestkin, Olga Svarnik

**Affiliations:** 1Moscow Institute of Physics and Technology, 1 “A” Kerchenskaya St., 117303 Moscow, Russia; olgasvarnik@gmail.com; 2Institute of Psychology of Russian Academy of Sciences, 13 Yaroslavskaya St., 129366 Moscow, Russia

**Keywords:** brain, learning, adaptation, hippocampus, information processing, integrated information theory, entropy

## Abstract

The amount of integrated information, Φ, proposed in an integrated information theory (IIT) is useful to describe the degree of brain adaptation to the environment. However, its computation cannot be precisely performed for a reasonable time for time-series spike data collected from a large count of neurons.. Therefore, Φ was only used to describe averaged activity of a big group of neurons, and the behavior of small non-brain systems. In this study, we reported on ways for fast and precise Φ calculation using different approximation methods for Φ calculation in neural spike data, and checked the capability of Φ to describe a degree of adaptation in brain neural networks. We show that during instrumental learning sessions, all applied approximation methods reflect temporal trends of Φ in the rat hippocampus. The value of Φ is positively correlated with the number of successful acts performed by a rat. We also show that only one subgroup of neurons modulates their Φ during learning. The obtained results pave the way for application of Φ to investigate plasticity in the brain during the acquisition of new tasks.

## 1. Introduction

An integrated information theory (IIT) and its main concept—the amount of integrated information Φ—were initially proposed, and successfully applied to quantify a level of consciousness in the brain [1,2,3,4,5,6,7], in nervous systems in simpler organisms [8], and in computational models [9]. This index increases in the conscious states of the brain, and is relatively small in unconscious states. At the same time, since this theory was released, it was shown that Φ reflects the performance of many non-brain, complex, self-organized systems consisting of some interacting nodes [10,11,12,13] quite well. It can be used to predict the level of success in a task, expressed in standard terms of an environment where the system operates. Researchers claim that Φ reflects the internal state of a self-organized system (“what the system is”), instead of describing externally visible system features (“what the system does”) [12]. There is evidence that Φ depicts the important internal state transitions in a complex system [14]. It describes the dynamical complexity of a system [13,15], rising from the local information exchange between self-organizing elements. This is applicable for all complex systems with nodes that can be in “on” and “off” states. For this reason, we think investigating the relationship between phi and the performance of a brain performing a task will provide novel results. An integrated information theory was applied to a brain model with neural network learned using the Hebbian rule, and the simulated EEG based on this network. The learning progress led to the incremental increase in Φ, and Φ exhibits a correlation with a number of attractors in a network [16]. Calculated from an EEG recorded in human infants, Φ increases with their age, which shows that Φ reflects the development of the brain. [17].

The concept of Φ is based on the quantification of two abilities of a system—a generation (segregation) of information, and its integration. First, a system generates as much information as possible. In more detail, a system has a maximal repertoire of possible states. This allows a system to distinguish more external events or stimuli, and change a state individually for each of them. Different parts of a system are specialized in the processing of different features (for example, different objects in an environment). Second, a system integrates the information. It means that the states of system components are interdependent, which is crucial to make inferences from information encoded in different components [4,10]. To act in an environment, a system makes a decision based on the available information. Without any integration, the information encoded by a variety of states of a system is useless, because it cannot be exploited by other parts of the system. For example, the visual system in the human brain integrates information about shape and color, and identifies them as a known object, such as a house, a tree, or a car. The coefficient Φ quantifies the degree of such integration. Small values of Φ means that there is a lack of connectivity in a certain place in a system, and the information transfer in this place is hampered, which, in turn, lessens the performance of a system.

According to the original definition, Φ is calculated as:(1)$$\Phi =\overset{r}{\underset{k=1}{\sum }}H(M_{t,k}|M_{t+\Delta t,k})-H\left(X_{t}|X_{t+\Delta t}\right)$$
where H(A|B) is the conditional entropy of variable A with given knowledge of variable B; $X_{t}$ and $X_{t+\Delta t}$ are the system state vectors at time moments t and t+Δt (see later); $M_{t,k}$ and $M_{t+\Delta t,k}$ are the state vectors of system subsets at time moments t and t+Δt; and r is the number of subsets. The expression written here is the information loss caused by the separation of a system into independent subsets $M_{k}$, breaking all connections between these subsets. Time here is discrete; it is a set of sequential system states taken at the same time intervals. Partition by subsets, so-called minimum information partition (MIP), is performed in such a way that subsets are maximally independent, i.e., this partition minimizes Φ. This rule highlights a mostly weak spot of a system, and prevents its coverage by any other well-interacting parts. In the extreme case, when some absolutely independent parts (without any connections between them) are present in a system, Φ=0, which depicts the full absence of information integration between these parts. The higher Φ is, the harder it is to split a system into parts without breaking any connections, and the more interconnections that are present between all parts of a system. This decreases the length of the path of information transfer, allowing the system performance to increase. Therefore, Φ is a good metric for the quantification of the performance of a system.

However, there is a problem in its practical application. A calculation of Φ is computationally challenging, due to the definition required to obtain a MIP. It can be found only using the brute force method, by checking all possible partitions of a set of neurons, and estimating the loss of information due to every partitioning, as in (1). The total number of such partitions in a network of N neurons is expressed with a Bell number $B_{N}$ [18], which grows faster than an exponent. For example, $B_{10}=115,975$ and $B_{25}=4,638,590,332,229,999,353$. This makes it practically impossible to calculate Φ for a real dataset. Moreover, the estimation of a conditional entropy in (1) requires a normal distribution of input data, which is not necessary for real neural time-series spike data. A limited set of a neural system states registered during an experiment may lead to the poor estimation of data distribution, which causes an instability of Φ [19]. Thus, calculations of Φ for neural time-series data require another approach. One of the most widely used algorithms for such data is the so-called autoregressive Φ ($\Phi_{AR}$), proposed by Barrett and Seth. One can write [19]:$$\Phi_{AR}=\frac{\underset{M}{\min}\left(\frac{1}{2}\ln\frac{\det\left(\Sigma \left(X\right)\right)}{\det\left(\Sigma \left(E^{X}\right)\right)}-\sum_{k=1}^{2}\frac{1}{2}\ln\frac{\det\left(\Sigma \left(M_{k}\right)\right)}{\det\left(\Sigma \left(E^{M_{k}}\right)\right)}\right)}{L\left(M\right)}$$
where Σ(X) is the covariation matrix of variable X (the same system state vector as in (1); $E^{X}$ is the autoregression residuals (errors of regression that predicts the value of *X* at time moment t+Δt based on value of *X* at time moment *t*); $M_{k}$, k∈{0,1} are the two system subsets; $E^{M_{k}}$ is the autoregression residuals of variable only containing the states of elements contained in a subset $M_{k}$, and $L\left(M\right)=\frac{1}{2}\ln\left[\underset{k}{\min}\left\{{\left(2\pi e\right)}^{\left|M_{k}\right|}\det\left(\Sigma \left(M_{k}\right)\right)\right\}\right]$ is the normalization factor (here $\left|M_{k}\right|$ is a size of a subset $M_{k}$). Normalization is used to diminish the influence of subsets sizes—for example, if a system is split into a big part and a small part, non-normalized Φ for such a partition is almost less than in case of partitioning into equal parts. Subsets are selected in a way to minimize the resulting $\Phi_{AR}$ value. If the input data have a normal distribution, $\Phi_{AR}$ calculated in this way is equal to Φ calculated using a classical definition [19]. Here, the simplification is made—the minimum information partition is replaced by the minimum information bipartition (MIB). Thus, only two subsets of a system need to be estimated. The number of candidate bipartitions is defined by a Stirling number of the second kind, $S_{N}^{\left(2\right)}=1/2\left(2^{n}-1\right)$ [20], for example, $S_{10}^{\left(2\right)}=511$ and $S_{25}^{\left(2\right)}=16,777,215$, which are significantly less than corresponding numbers for classic Φ definition. But this is still a very time-consuming task, with calculation time growing exponentially. We experimentally estimated an upper limit of 15 neurons on a modern computer (AMD Ryzen 7 4800H 2.9 GHz, 16 GB 3.2 GHz DDR4 RAM).

In this study, we compared different approximation methods of Φ calculation, which made it possible to accurately calculate Φ for a reasonable time, and applied them to the investigation of plastic changes in a rat brain during an instrumental hippocampus-dependent learning in a maze. We performed calculations using the autoregressive Φ algorithm on a limited set of neurons, using a bipartition calculated with a brute force method, and on a full set using two approximate partitions. We investigated a correlation of Φ with an environment-specific metric—the number of successfully performed acts. The obtained results show the applicability of all approximations used for the calculation of Φ, and its ability to predict a learning progress.

## 2. Materials and Methods

### 2.1. Behavioral Experiment

For this study, we used a public dataset from crcns.org, with data recorded from a hippocampus [21,22,23]. Hippocampus-dependent learning is one of the most frequently and actively used kinds of behavioral experiments with the acquisition a new task. Therefore, all protocols of such experiments are well-known and well-developed, and a lot of patterns are known in the activity of hippocampus neurons, during all stages of learning. That was the reason for selecting such dataset in our calculations. Rats (*n* = 3, Long–Evans) were set a complicated task in an W-shaped maze, with liquid reward sites at the ends of each arm. To obtain a reward, rats were required to make a full run from an end of one arm to the end of another arm in accordance with some rules (described in original manuscript [21]). We selected a dataset with such a complicated task to ensure that the task led to a significant cognitive workload in the brain and, thus, according to our hypothesis, caused a large change of Φ. After acclimatization to the maze, an array of microelectrodes were implanted into the right hippocampus, CA1 region (coordinates: −3.6 mm AP, 2.2 mm ML). The registration of action potentials using implanted microelectrodes gave the best time resolution, because it directly measured a voltage on a neuron membrane. Time patterns of spike activation encode information, so it is crucial for information-theoretic methods to register as precise spike times as is possible. Learning started after a rehabilitation period of 5–7 days. Each day, one or two experimental sessions were performed for 15–20 min. Between 15 to 50 neurons were registered during each session.

### 2.2. Applied Approximation Methods for Φ Calculation

Taking only some small part of neurons is a reasonable approximation itself. Such simplifications shorten the time of brute force in search of a minimum information bipartition, which is usually required in the autoregressive Φ algorithm. This approximation was used in [11] for a non-brain complex system. A limited dataset was suitable for the calculation of $\Phi_{AR}$, and this value reflected a state of a system. For a brain network, the hypothesis was made that a small subset of neurons reflects the state of the whole brain subregion. The brain is a multi-level network of neurons with structured patterns (interconnected dense hubs) observable at each level [24,25], and probing a small group of neurons, gives an approximate picture of the whole region. Existing registration methods, with a single-neuron resolution, take only a part of a brain, up to some hundreds of neurons (for example, [26,27]), and this is still only a small piece of the brain. As mentioned above, the upper computational limit is 15 neurons. In dataset used, the lowest registered number of neurons is 15, and the greatest is 56 (see Appendix A). Thus, this approximation yields an information loss from 0% (at best), to 73% (at worst).

The other way to avoid such information loss is by finding a system partition on the whole set of neurons, but using an approximate way. One of the methods exploits the mathematical properties of the MIP as a function to find its minimum, meaning the MIP is found with a reduced number of considered partitions [28] Other approximations of this class are based on replacing a minimum information partition with any other, approximately reflecting its essence. This allows the calculation of Φ considering only one partition. A rough approximation is so-called atomic partition [10,29]. A system of N neurons is split into N subsets, where each subset contains one neuron (Figure 1b). This method considers only single neurons as computational units, and neglects information generation by neuron clusters. A structure of the brain is represented by densely connected neuron groups, which, in turn, are connected to each other with a subset of neurons with many synapses (the so-called rich club). Consequently, this approximation ignores brain structure and information properties arising from such a structure. But this approximation is the fastest to calculate and, thus, we tested it to check the borders of its application. We used a modified autoregressive Φ ($\Phi_{AR}$) algorithm with an atomic partition. For more precise calculation, other approximations, that take network structure into account, are required.

As noted earlier, this partition is required to highlight weakly interdependent parts of a system, which yield to the reduction in information integration. More sophisticated approximation algorithms are based on a premise that the least interdependent parts are the least physically interconnected parts [30] Dense networks mentioned earlier, described by the “Small-World” model, are thought to support information segregation and specialization, and hubs connect such communities with each other, and perform information integration [31] (Figure 1a). Thus, the detection of such dense communities yields a partition into subsets performing information segregation (Figure 1c). A synaptic structure of a neural network is easily defined using the correlations between spikes in time-series data. Then, community detection algorithms, for example the Louvain algorithm [32], spectral clustering [33], or Weight Stochastic Block model [34] are used. This yields a relatively fine, approximated calculation of Φ.

To obtain such community structure, a correlation matrix between time-series data of all registered neurons was calculated. For each pair of spike trains of the neurons *i* and *j*, a correlation coefficient was calculated, and the null hypothesis that there is no relationship between these spike trains was tested. Each cell $A_{ij}$ of a final correlation matrix A contained a correlation coefficient in a corresponding element in the case of a relationship between spike trains i and j (if p<0.05), and zero otherwise. Such a matrix is interpreted as a network connectivity matrix, where a cell $A_{ij}$ shows a strength of connectivity between nodes i and j. Then, we used a Louvain algorithm to split a neural network into dense communities. Finally, $\Phi_{AR}$ was calculated with an estimated Louvain partition.

The Louvain algorithm is a widely known method for the detection of dense communities in a graph where each edge between nodes i and j has a strength (weight) $A_{ij}$. The algorithm is based on a maximization of a special metric M, called the modularity [32]:$$M=\frac{1}{2m}\underset{i,\;j}{\sum }\left(A_{ij}-\frac{k_{i}k_{j}}{2m}\right)\delta \left(c_{i},\;c_{j}\right),$$
where $m=\sum_{i,j}A_{ij}$ is the sum of weights of all edges in a graph; $k_{i}=\sum_{j}A_{ij}$ is the degree of node i; $c_{i}$ is the community of which a node i is a member of; and δ(x,y) is a Kronecker delta function: δ(x,y)=1 if x=y and δ(x,y)=0, respectively.. In this case, $\delta \left(c_{i},c_{j}\right)=1$ when the nodes i and j are in the same community, and $\left(c_{i},c_{j}\right)=0$. Initially, each node is assigned to its own community (similar to the atomic partition). Then, at each step, an algorithm tries to move each node i into a community each containing its neighbor. For each such attempt, a modularity change ΔM is calculated:$$\Delta M=\left(\frac{\sum_{in}+2k_{i,in}}{2m}-{\left(\frac{\sum_{tot}+k_{i}}{2m}\right)}^{2}\right)-\left(\frac{\sum_{in}\;}{2m}-{\left(\frac{\sum_{tot}}{2m}\right)}^{2}-{\left(\frac{k_{i}}{2m}\right)}^{2}\right),$$
where $\sum_{in}$ is the sum of weights of all edges inside the new community (the community where i is moving into); $k_{i,in}$ is the sum of weights of edges connecting the node i with all elements of the new community; and $\sum_{tot}$ is the sum of external links to nodes of the new community (sum of links to nodes contained in a new community from nodes not contained in a new community). Finally, a node i is moved into the community yielding the biggest value of ΔM. This step is repeated until the maximal modularity is achieved. Finally, all nodes (neurons) cluster into communities based on their connectivity.

### 2.3. Calculation of Φ for Time-Series Spike Data

To calculate $\Phi_{AR}$ using neural data, nodes of a network and their states at each time moment must be defined. Obviously, nodes are the neurons, and states of nodes are defined in the following way [35]. A time is split into equal intervals (bins). If a neuron produced a spike in a bin, an “on” state is assigned to this bin, and vice versa. A bin size is defined as a compromise between keeping all fine neuron spiking patterns, and storing excessive information that slows down computations. Moreover, too small a bin size may cause an algorithm to fail, because most bins will have a zero state. According to these rules, we empirically defined a bin size as 0.03 sec. Calculations were performed using a MATLAB script, with the use of a modified open-source code by Barrett and Seth [19]. Calculation of a maximum modularity partition (MMP) for an MMP approximation, using a Louvain algorithm, was performed using an open-source MATLAB library GenLouvain [36].

For some periods, an algorithm was unable to compute the value of $\Phi_{AR}$. This happens when an input sequence of states of any neuron is too homogenous, i.e., a neuron was almost always in an “on” or “off” state. In such circumstances, matrices in autoregressions in (2) become rank-deficient, and an algorithm is unable to return a value [11]. In this case, a neuron with the lowest Bernoulli variation of activity was removed, and the computation repeated. It was performed until the valid value of $\Phi_{AR}$ was obtained. If all bins in the time-series of a neuron are in the “off” or “on” state, the Bernoulli variance is equal to zero. On the contrary, if the numbers of “on” and “off” states are equal, the Bernoulli variance is equal to 1/4, and this is the maximal value. The same method of neuron selection was also used to obtain 15 neurons in the first (15-neuron) approximation. This metric does not reflect the order of bins and, hence, misses long sequences of “off” and “on” states, which obviously decrease the variance. However, such long “on” and “off” periods may encode important information, for example, some place-specific cells in a hippocampus are only active in particular place in a space [37]. Moreover, this criterion is simple for calculation, and was useful in our computations, with many experimental sessions present, and the selection of neurons performed for each of them.

For the 15-neuron approximation, a subgroup of 15 neurons was selected using the same criterion—neurons were sorted using the Bernoulli variance, and 15 neurons from the top of that list were used. In other methods, all registered neurons were used to perform calculations, except for ones removed due to the impossibility of $\Phi_{AR}$ calculation, as mentioned above.

A definition of Φ contains a parameter Δt, which shows a time interval between two sequential states of a system.. This value is a characteristic property, which reflects the period of main information processing in a system. In the brain, many processes of different time scales take place, such as fast processes of spiking, or slow processes in large networks of neurons. As in previous findings, the processes of attention, conscious perception, and information processing are periodic [38,39,40], and the period, due to different points of view, lasts from about 50 to 500 ms [17]. Just as in [11], we took Δt, which yielded the greatest average value of $\Phi_{AR}$ for 8 days. This method allows for finding times that are mostly valuable information-handling processes in the brain.

### 2.4. Statistical Analysis

The integrated information coefficient $\Phi_{AR}$ was calculated for each learning session. For days when two learning sessions were conducted, a resulting value of $\Phi_{AR}$ was calculated, as an average between $\Phi_{AR}$ values during this day. To obtain a bigger resolution, and capture thin patterns in Φ dynamics, each session was divided into 8 periods of equal length, and Φ was calculated for neuron activity in each period. Each calculated value of $\Phi_{AR}$ was normalized by the number of neurons used in a calculation to provide the possibility of comparing such values. The ability of information segregation and integration depends particularly on system size; a small system will have a smaller repertoire of possible states and generate a smaller amount of information. Thus, to eliminate this effect, normalization was performed. The number of successfully performed acts, mean neuron frequency, and mean interspike interval were also calculated for these periods. Learning session lengths were slightly different, so the number of rewards was normalized using the length of periods. Finally, correlations between $\Phi_{AR}$ and relative number of successful acts were estimated, and a null hypotheses about absence of correlation was tested. For the measurement of a correlation, a Spearman correlation coefficient was used. The null hypothesis was tested using a Student distribution. Bonferroni corrections for multiple comparisons were applied; for 15 tested hypotheses, in order to be significant, the *p*-value must be less than α=0.05/15=0.0034.

## 3. Results

All animals successfully acquired a task, and exhibit progress during the whole experiment time. Learning curves are found in Appendix A.

The integrated information coefficient $\Phi_{AR}$ was calculated using all three approximation methods in all animals. The results for each day are shown in Figure 2. The Δt determined for each condition during optimal information-processing periods are shown in Table 1. The obtained periods are of a scale comparable with earlier described time scales of periodic attention and information processing (about 0.5 s) [17].

The $\Phi_{AR}$ calculated on the full set of neurons using approximated partition (Louvain and atomic) are almost equal. An atomic partition gives the result with a maximal magnitude, which is simply explained. A Louvain partition quantifies a network more rigorously, and reveals a greater number of weak spots in a system. The atomic partition only considers the restricted part of network partitions, and can miss lack of interaction between some parts of a system. Thus, the atomic partition yields larger values of Φ . The $\Phi_{AR}$ calculated using a 15-neuron approximation differs strongly from that calculated with other approximations, but it catches the main trends of their temporal change, in spite of some points. This value is one order of magnitude less than $\Phi_{AR}$ obtained with other approximations, which can be explained by a number of neurons used in a calculation. Obviously, 15-neuron networks are less complicated compared to those with more neurons, thus, the amount of information integration in 15-neuron networks is less than in larger networks. This yields a lesser value of $\Phi_{AR}$ despite the normalization to the number of neurons.

For the 15-neuron approximation, in almost all sessions $\Phi_{AR}$ was successfully calculated on the whole set of 15 neurons, without rank-deficient matrices. For partition approximations, in a valuable fraction of 33 sessions, a limited subset of all registered neurons was used. But these approximations demonstrate a huge benefit in the number of considered neurons, thus, avoiding the loss of information encoded in these neurons. The most benefit is observed in the second animal; $\Phi_{AR}$ is successfully calculated for 55 neurons (day 3, session 2, period 3), instead of 15 neurons in the first approximation. (See the number of registered neurons in the Appendix A). The number of neurons successfully used to perform calculations is shown in Appendix A.

The results of correlation analysis of different approximations of $\Phi_{AR}$, and a relative number of rewards, are shown in Table 2.

A 15-neuron approximation yields good results: an observable metric of learning success (a relative number of rewards) is significantly correlated with the integrated information coefficient $\Phi_{AR}$. Other approximations give weaker results; $\Phi_{AR}$ calculated with these approximations does not show a significant correlation with metrics of learning, in some cases. Scatter plots demonstrating this correlation are shown in Figure 3.

Based on these findings, we hypothesize that a problem with the Louvain and atomic approximations arises from the additional neurons used in these methods, but removed in the 15-neuron approximation. Such neurons are presumably not connected with a main network associated with execution of the main task, and in such a case, their integration degree is not correlated with a task execution. This group of neurons changes the total Φ and, thus, destroys a correlation. In support of this hypothesis, it is noted that for the second animal, about 40–50 neurons were used in calculations; which is more than 20–30 neurons for the first animal. Thus, the second rat contained almost twice as many additional neurons that were used in the calculation of Φ with the Louvain and atomic approximations. To check this hypothesis, we kept only 15 neurons, using the same algorithm used for the 15-neuron approximation (with the maximal Bernoulli variance), and calculated $\Phi_{AR}$ for this limited set of 15 neurons, while the partitions were the Louvain and atomic partitions. Thus, the same method with autoregressive Φ and approximate partition was applied, but only a small subset of neurons was used. The results are shown in Figure 4 and Table 3. This approach yields a good, statistically significant correlation. It confirms a hypothesis about excessive neurons not associated with a targeting task, and not changing their degree of information integration during learning. The removal of these neurons yields a good correlation of $\Phi_{AR}$, with an index of learning success.

## 4. Discussion

We investigated time changes in the integrated information coefficient ($\Phi_{AR}$) for neural activity during a process of complex instrumental learning, and its correlation with known environment-dependent markers of learning. The number of successfully performed acts is an environment-specific and task-specific metric of learning progress and, thus, is not universal. We show the presence of a cohort of neurons that change their degree of information integration during learning. The successful execution of a task is dependent on the degree of information integration of this neuron subgroup. Of course, contemporary registration methods allow the registering of only a small part of this network, and cannot even give a guarantee that the same neurons from this cohort are registered during the process of learning. However, we show that the integrated information theory successfully and correctly describes the performance of a neural network, even only having the small subset. This builds up a picture of a neural network performance from these fragments. Thus, the integrated information coefficient is a marker of neuroplasticity in different brain regions during the acquisition of new memory.

We compared different methods of approximated calculation of $\Phi_{AR}$. Louvain and atomic partition were successfully applied for spikes data, and overcame the technical limit of 15 neurons achievable on modern computers. For this relatively small dataset, the number was excessive, because only a fraction of registered neurons is associated with execution of a task. Other neurons did not demonstrate a variation of information integration related with the execution of a task. Thus, a 15-neuron approximation is the best for experiments with the same neuron number as this dataset, because a probability of capturing a large number of task-related neurons is very small.

In previous works investigating the integrated information in a brain, the focus of attention is directed to the EEG data—simulated [16], and in vivo in a developing brain [17]. We investigated a possibility of utilization of neural spike data recorded from the individual cells. This improvement significantly helps in the investigation of plastic changes in a brain, because it paves the way to deal with individual cells encoding and processing information, and not with data averaged from multiple neurons, as in the EEG method. This also allows experiments to be performed involving free-behaving animals. We made this calculation with different partitions, extending and generalizing the calculation algorithm. For time-series data, a widely used autoregressive algorithm was applied [19], as in other research with time-series data [17]. Barrett and Seth, in their work about an autoregressive algorithm, make a significant simplification, and limit their work to system bipartitions, i.e., partitions of a system into two parts. In two approximations (atomic and Louvain), we overcame this limit and replaced a bipartition with a partition approximately describing the structure and information processing in a system. We investigated the correlation between Φ and the classical, observable, and task-dependent metric of success (the number of rewards), as performed earlier in non-biological complex systems [10,11]. All previous studies show a trend of Φ increasing during learning [10,17], and/or a correlation between Φ and a system performance metric [10,11,16,17]. In our work, only the second is observed. The rising trend of Φ is not observed. The possible reason for such behavior is that not all of the whole set of neurons involved in task performance were registered, and this part does not change Φ significantly to obtain the trend.

Additional experiments are required to know more about the self-organization of brain networks and their evolution. New experiments must involve more animals learning different tasks, of varying complexity. It would be interesting to investigate the modulation of Φ for different tasks, and check whether the complexity of a task is related to the degree of Φ variability. One more interesting hypothesis is the relation between Φ and the metric of the performance of each individual rat, e.g., the learning speed or the maximal number of rewards achieved during learning. Of these three rats, the second rat demonstrates the minimal number of rewards and a minimal learning speed, and its value of $\Phi_{AR}$ has the minimal magnitude across all rats. The opposite is also true—the third rat finally achieved the greatest number of rewards, and its value of $\Phi_{AR}$ was the maximal. It is also interesting to try to predict the final success of an animal using $\Phi_{AR}$ on the first day only. If this hypothesis is confirmed, it is possible to predict the success at very early stages. More complicated methods for neural activity registration that recently appeared and may capture a big neuron ensembles, and it will be possible to highlight a big group associated with task performance. It will be useful to test approximations such as the Louvain or atomic partition on this neural group, and test how it reflects changes in a brain during learning. The next stage in investigation is human experiments. There are ways to calculate Φ on EEG or fMRI data [16,17], and experiments with calculation of $\Phi_{AR}$ in humans performing different tasks, or even in long, multiple-day learning can produce a lot of new information about the neural basis of complex behavior, inherent only to humans.

## 5. Conclusions

Our results confirm the possibility of the integrated information coefficient (Φ) for real-time investigation of self-organized neural networks in the brain. This ability was proposed by previous studies, but the calculation of Φ for time-series spike data was computationally challenging and, thus, not possible. We applied three different approximation methods to spike data from the hippocampus in rats acquiring a new task, and the Φ obtained by three methods successfully predicts the learning success. Moreover, we show that a subgroup of neurons does not participate in the execution of a task. These results are useful in the application of Φ as an environment-independent and task-independent metric of brain adaptation to new environments.

## Figures and Tables

**Figure 1 brainsci-12-00596-f001:**
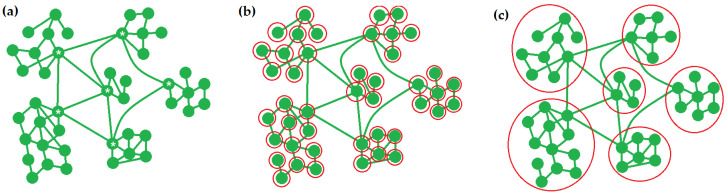
A community structure in a brain neural network and partitions based on this structure and reflecting information processing. (**a**) Dense community and hub structure. Six communities with densely interconnected neurons connected with each other by a cohort of neurons, called hubs (marked with asterisks). Communities are responsible for information segregation and specialization, and hubs are responsible for information integration between these communities. (**b**) Atomic partition. Each neuron is included in its’ own subset (marked with red circle). This partition does not reflect information segregation and integration processing. (**c**) Partition based on community detection algorithms. This partition separates communities performing information segregation, and is close to the desired minimal information partition (MIP).

**Figure 2 brainsci-12-00596-f002:**
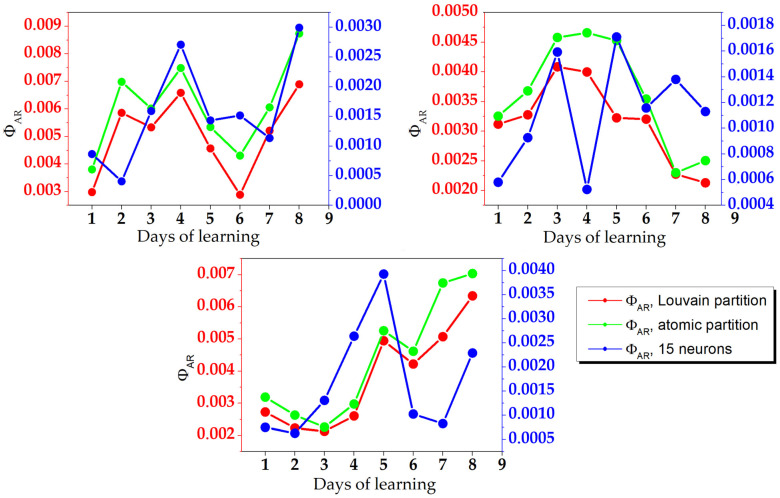
$\Phi_{AR}$ calculated for three animals for each learning day, using three different approximation methods. Left y axis: Louvain partition and atomic partition approximation. Right y axis: 15-neuron approximation.

**Figure 3 brainsci-12-00596-f003:**
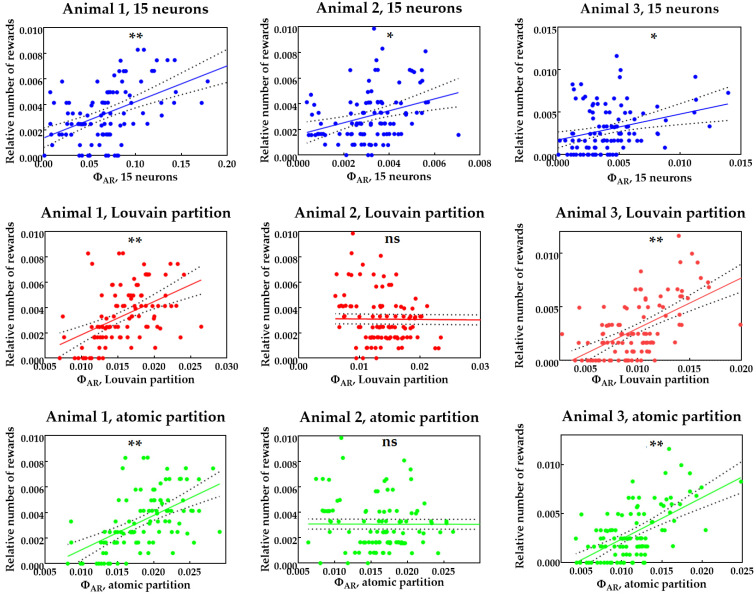
$\Phi_{AR}$ for each period of 1/8 of learning session vs. relative number of rewards for each rat and each method of approximation. Significance levels are shown according to Bonferroni correction for multiple comparisons (15 checked hypotheses totally): “ns” means the absence of significant correlation (*p* > 0.05/15 = 0.0034), “*” means p<0.05/15=0.0034; “**” means $p<0.01/15=6.667\times 10^{-4}$.

**Figure 4 brainsci-12-00596-f004:**
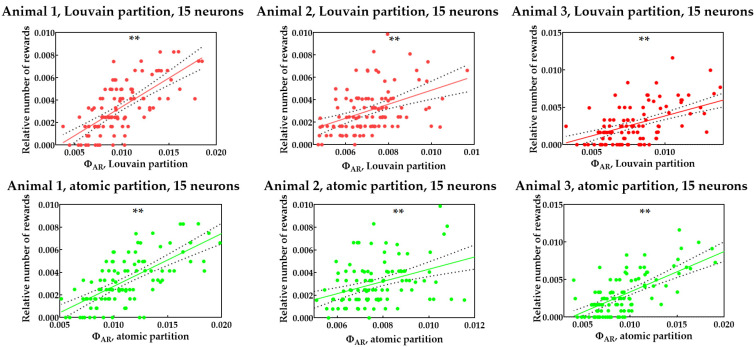
$\Phi_{AR}$ vs. relative number of rewards for each rat in periods, with $\Phi_{AR}$ calculated using approximations with Louvain and atomic partitions on the limited set of 15 neurons. Significance levels are shown according to Bonferroni correction for multiple comparisons (15 checked hypotheses totally): “ns” means the absence of significant correlation (*p* > 0.05/15 = 0.0034), “*” means p<0.05/15=0.0034; “**” means $p<0.01/15=6.667\times 10^{-4}$.

**Table 1 brainsci-12-00596-t001:** Optimal periods of information processing for three animals, obtained by maximization of Φ calculated with three different approximations.

	15 Neurons	Louvain Partition	Atomic Partition
Animal 1	14 bins (0.42 s)	15 bins (0.45 s)	15 bins (0.45 s)
Animal 2	16 bins (0.48 s)	17 bins (0.56 s)	13 bins (0.39 s)
Animal 3	13 bins (0.39 s)	12 bins (0.36 s)	12 bins (0.36 s)

**Table 2 brainsci-12-00596-t002:** Results of correlation analysis of a relative number of rewards as a well-known marker of learning success, and integrated information coefficient ($\Phi_{AR}$) calculated using three different approximations. r is the Spearman correlation value. Significance levels are shown according to Bonferroni correction for multiple comparisons (15 checked hypotheses totally): “ns” means the absence of significant correlation (*p* > 0.05/15 = 0.0034), “*” means p<0.05/15=0.0034; “**” means $p<0.01/15=6.667\times 10^{-4}$.

	15 Neurons	Louvain Partition	Atomic Partition
Animal 1	r = 0.4796 *p* < 0.0001 **	r = 0.5364 *p* < 0.0001 **	r = 0.5911 *p* < 0.0001 **
Animal 2	r = 0.3229 *p* = 0.0008 *	r = −0.2274 *p* = 0.0213 ns	r = −0.1293 *p* = 0.3987 ns
Animal 3	r = 0.3096 *p* = 0.0014 *	r = 0.5726 *p* < 0.0001 **	r = 0.6544 *p* < 0.0001 **

**Table 3 brainsci-12-00596-t003:** Results of correlation analysis of a relative number of rewards, and integrated information coefficient ($\Phi_{AR}$) calculated using approximations with Louvain and atomic partitions on the limited set of 15 neurons. Significance levels are shown according to Bonferroni correction for multiple comparisons (15 checked hypotheses totally): “ns” means the absence of significant correlation (*p* > 0.05/15 = 0.0034), “*” means p<0.05/15=0.0034; “**” means $p<0.01/15=6.667\times 10^{-4}$.

	Louvain Partition	Atomic Partition
Animal 1	r = 0.6767 *p* < 0.0001 **	r = 0.6645 *p* < 0.0001 **
Animal 2	r = 0.4557 *p* < 0.0001 **	r = 0.3952 *p* < 0.0001 **
Animal 3	r = 0.6060 *p* < 0.0001 **	r = 0.6544 *p* < 0.0001 **

## Data Availability

An open dataset from crcns.org was used. This dataset can be found at: https://crcns.org/data-sets/hc/hc-28/about-hc-28 (accessed on 1 May 2022).

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
