# Peer review of "Different Approximation Methods for Calculation of Integrated Information Coefficient in the Brain during Instrumental Learning"

_brainsci, 2022, doi:10.3390/brainsci12050596_

Round 1

Reviewer 1 Report

Report

The authors compare different approximation methods for the calculation of integrated information coefficient in the brain neural networks during instrumental learning.

This study is interesting and should be considered for publication as long as the following comments are taken into account

Comments:

There are some typos in the manuscript, that must be corrected, pass the article through a linguistic corrector

It is not clear from the article how the multi-agent networks used for the calculations have been selected. Please include a detailed explanation in the text.

The images need to be improved, they are small and do not display properly.

Author Response

>There are some typos in the manuscript, that must be corrected, pass the article through a linguistic corrector

We have corrected mistakes, a native English speaker from the Stony Brook University, New York reviewed this paper.

>It is not clear from the article how the multi-agent networks used for the calculations have been selected. Please include a detailed explanation in the text.

An explanation about the selection of neurons was added (lines 289-293).

>The images need to be improved, they are small and do not display properly.

 Images were improved: bigger fonts were used for tick labels and axis/subplot titles; lines and markers were replaced by thicker ones, and the resolution of images was raised. Fonts were driven to accordance with the font used in the manuscript template.

Please find attached the corrected version of our manuscript.

Reviewer 2 Report

Dear authors,

I appreciate your efforts in writing this article. There are some mistake in article that need to revise. Also, some of the important points of the article should be clearly stated.

1- Many grammatical errors in the paper needs to revise.
2- After equation 1, the authors introduce P(A), P(B), and p(A, B). why?
3- It can be beneficial that the results are compared with other papers, especially the reference [14].

4- General, it is useful that the mathematical details of the methods are also discussed.

Best regards

Author Response

1- Many grammatical errors in the paper needs to revise.

We have corrected the English grammar in our manuscript. It was reviewed by a native English speaker from from Stony Brook University, New York.

2- After equation 1, the authors introduce P(A), P(B), and p(A, B). why?

These unused definitions were removed (lines 81-82). Originally, we introduced a definition of a conditional entropy, and P(A), P(B), and p(A, B) were contained in such definitions. Later, we decided that the definition of a conditional entropy is trivial and does not need to be written in the text, and, unfortunately, forgot to remove these remaining definitions. We apologize for this mistake due to inattentiveness.

3- It can be beneficial that the results are compared with other papers, especially the reference [14].

 We added a paragraph comparing our research with other ones (lines 434-455), including a comparison of used Autoregressive algorithm from ref. [14] with algorithms used in a manuscript in terms of used partition.

4- General, it is useful that the mathematical details of the methods are also discussed.

We added some clarifications into descriptions of used mathematical methods: of the Autoregressive  algorithm (line 117-118, 120-121, 123-125), of the 15-neuron approximation (lines 173-175, 289-291) and of the Louvain partition (lines 251-252).

Please find attached a revised version of a manuscript.

Reviewer 3 Report

In this paper, author (s) compared different approximation methods for Φ calculation in neural spike data and investigated how does it describe a degree of adaptation in brain neural networks. They showed that in the rat hippocampus during an instrumental learning, despite some amplitude discrepancies, all applied approximation methods reflect temporal trends of Ф, and a value of Ф is positively correlated with a number of successful acts performed by a rat. However, we have some comment to improve the quality of the paper as follows:

  1. Please modify the abstract to describe the novelty and contribution of the paper.
  2. Introduction needs further clarifications.
  3. In section 2, How to motivate to consider the method.
  4. Please improve the quality of figures.
  5. Result and discussion section are in good shapes.
  6. I suggest to author(s) that please add the conclusion section to summarize the study.
  7. Author(s) cited 34 references in the manuscript, but most of the references out of dated. Please cite some latest references from 2020-2022 to paper.

Author Response

>Please modify the abstract to describe the novelty and contribution of the paper.

The abstract was modified (lines 9-29). The actual problem with a computation was highlighted, some results clarified and the practical application of reported methods was specified.

>Introduction needs further clarifications.

We added more detailed information about the concept of Φ: an exact information what Φ describe a degree of a complexity in complex systems (lines 46-48); wrote that systems are the self-organizing systems of interacting elements (line 44); extended claims about information segregation and integration, about what they are and why are they important for a performance of a complex system (lines 61, 64-66, 71-74); . We also described how does the value of Φ may be interpreted and what processes in a system does it reflect (lines 74-77). We added information about the essence of mathematical expression in the definition of Φ (lines 85-87), and about the interpretation of a calculated Φ value (lines 91-97). We removed a sentence breaking a logic of introduction (lines 58-59). We extended a description of results obtained in cited manuscripts (lines 53-58). The paragraph describing a calculation of  was ended with a sentence describing the motivation of  calculation (lines 97-98).

In section 2, How to motivate to consider the method.

If you are writing about the method of spike data registration (implanted microelectrodes), we added its motivation (lines 162-166). If you are writing about each approximation method, a motivation is written in the manuscript text. The 15-neuron approximation was already successfully applied, as it written in line 175, ref. 9. The atomic partition was already used, which is mentioned in lines 193-194 (references 8, 23). The shortcomings of this method are described later (lines 195-200). The Louvain (clustering) partition is based on the synaptic structure underlying the information processing in a brain, which is described in lines 204-210.

Please improve the quality of figures.

Figures were improved: bigger fonts were used for tick labels and axis/subplot titles; lines and markers were replaced by thicker ones, and the resolution of images was raised. Fonts were driven to accordance with the font used in the manuscript template.

Result and discussion section are in good shapes.

Thank you.

I suggest to author(s) that please add the conclusion section to summarize the study.

The conclusion was added (lines 478-487).

Author(s) cited 34 references in the manuscript, but most of the references out of dated. Please cite some latest references from 2020-2022 to paper.

Some new references were added to a manuscript. They give a newest information about the integrated information coefficient  application to complex systems outside the framework of a consciousness. [35] additionally confirms that  well reflects an internal state of a complex system and its change is related with important transitions in a system. [36] shows an additional example of exploitation of  as a measure of a system complexity. [37] is the newest research confirming that Φ can be used as a metric of complexity.

Please find attached a corrected version of a manuscript.

Reviewer 4 Report

My comments are attached.

Author Response

1.To be more reader-friendly, some points can be explained more clearly like: the difference between the method using 15 neurons and the method using Atomic partitions on the limited set of 15 neurons.

We added some clarifications into the descriptions of applied methods: in lines 173-175 (method using 15 neurons) and lines 387-390 (method with Louvain/Atomic partitions and 15 neurons).

In the first method, the limited set of neurons was used to lessen the computational time (this method requires the brute-force search). In the second method, the limited set was used because of a hypothesis that additional neurons, usage of what in computation became possible due to approximation method, did not participate in execution of a task and thus lessen the  of the total system.

2.Some minor revision suggestions are:

1) On line 54 of page 2, the meaning of p(A), p(B), p (A, B) should be deleted since they are not used in the formula.

We removed them (lines 81-82). Originally, we introduced a definition of a conditional entropy, and P(A), P(B), and p(A, B) were contained in such definitions. Later, we decided that the definition of a conditional entropy is trivial and does not need to be written in the text, and, unfortunately, forgot to remove these remaining definitions.

2) On line 55 of page 2, “$X_{0}$ and $X_{1}$” should be “$X_{t}$ and $X_{t+\delta t}$”.

We fixed it (lines 82, 83). We apologize for this mistake due to inattentiveness.

3) On line 78 of page 2, the meaning of $E^{M_{k}}$ should be given here.

The meaning was added (page 120-121).

4) On line 149 of page 11, “and hubs connect … and are perform…” should be “and hubs connect … and perform…”.

This typo was fixed (now it is line 210).

5) On line 178 of page 4, “… Is the…” should be “… is the…”.

Fixed (now it is line 241).

6) On line 185 of page 4, the meaning of $k_{i, in}$ should be given here.

The meaning was added (lines 251-252).

7) On line 368 of page 10, the link of Figure S1 here can’t work.

 We have attached the Figure S1 in the file ‘manuscript-supplementary.zip’ when we were submitting the manuscript. I think it is a technical issue from the editorial office.

3.In addition, there are some related works about the integrated information and partition methods:

1) Tagliazucchi E. The signatures of conscious access and its phenomenology are cconsistent with large-scale brain communication at criticality[J]. Consciousness and cognition, 2017, 55: 136-147.

Thank you for this manuscript. There, the coefficient Φ was applied in a computational model and this confirmed the ability of  to reflect the conscious states of a system. We used it in Introduction as reference [38] (line 35) where we write about applications of Φ.

2) Engel D, Malone T W. Integrated information as a metric for group interaction[J]. PloS one, 2018, 13(10): e0205335.

This manuscript was already cited by us (ref. [9]) and we know it for a long time. Moreover, this manuscript inspired our research of Integrated information theory. It is very significant for us.

3) Kitazono J, Kanai R, Oizumi M. Efficient algorithms for searching the minimum information partition in integrated information theory[J]. Entropy, 2018, 20(3): 173.

This manuscript is useful too, because represents another class of algorithms of  calculation. We compared an algorithm from this manuscript with approximated algorithms used by us (ref. 41, lines 187-190).

Please find attached a corrected version of a manuscript.

Round 2

Reviewer 3 Report

The authors have revised very well in the revised version.